# Genome-Wide Association Studies for Lactation Performance in Buffaloes

**DOI:** 10.3390/genes16020163

**Published:** 2025-01-27

**Authors:** Wangchang Li, Henggang Li, Chunyan Yang, Haiying Zheng, Anqin Duan, Liqing Huang, Chao Feng, Xiaogan Yang, Jianghua Shang

**Affiliations:** 1Guangxi Key Laboratory of Buffalo Genetics, Reproduction and Breeding, Guangxi Buffalo Research Institute, Chinese Academy of Agricultural Sciences, Nanning 530001, China; kyzhang70@foxmail.com (H.L.); 1918401003@st.gxu.edu.cn (C.Y.); haiyingzheng@126.com (H.Z.); duanaq321@163.com (A.D.); 18269051851@163.com (L.H.); ylfchao@163.com (C.F.); 2Guangxi Key Laboratory of Animal Breeding, Disease Control and Prevention, College of Animal Science & Technology, Guangxi University, Nanning 530004, China; liwangchang1019@163.com; 3Key Laboratory of Buffalo Genetics, Breeding and Reproduction Technology, Ministry of Agriculture and Rural Affairs, Nanning 530001, China

**Keywords:** buffalo, milk production traits, genome-wide association study

## Abstract

**Background**: Buffaloes are considered an indispensable genetic resource for dairy production. However, improvements in lactation performance have been relatively limited. Advances in sequencing technology, combined with genome-wide association studies, have facilitated the breeding of high-quality buffalo. **Methods**: We conducted an integrated analysis of genomic sequencing data from 120 water buffalo, the high-quality water buffalo genome assembly designated as UOA_WB_1, and milk production traits, including 305-day milk yield (MY), peak milk yield (PM), total protein yield (PY), protein percentage (PP), fat percentage (FP), and total milk fat yield (FY). **Results**: The results identified 56 significant SNPs, and based on these markers, 54 candidate genes were selected. These candidate genes were significantly enriched in lactation-related pathways, such as the cAMP signaling pathway (*ABCC4*), TGF-β signaling pathway (*LEFTY2*), Wnt signaling pathway (*CAMK2D*), and metabolic pathways (*DGAT1*). **Conclusions**: These candidate genes (e.g., ABCC4, *LEFTY2*, *CAMK2D*, *DGAT1*) provide a substantial theoretical foundation for molecular breeding to enhance milk production in buffaloes.

## 1. Introduction

Water buffaloes hold immense significance in the dual domains of meat and milk production, with their milk making a considerable contribution to the global dairy sector. Specifically, buffalo milk accounts for an impressive share of over 15% of the global milk yield, underscoring the essential role of this species in the global dairy industry. This statistic highlights the importance of focusing on the welfare, productivity, and reproductive efficiency of buffaloes to ensure the continued growth and sustainability of this critical livestock sector [1]. Buffalo milk is distinguished by its elevated levels of fat, protein, and minerals when compared to cattle milk, which contributes to its nutritional superiority and economic value. This nutritional profile is a key factor in the greater prevalence of buffaloes in Asia, where they outnumber many other livestock species. The unique composition of buffalo milk not only enhances its suitability for dairy products but also reinforces the strategic importance of buffaloes in the agricultural economies of the region [2]. Furthermore, buffalo milk stands out in the market with economic significance, often fetching a price that is double that of bovine milk, reflecting its exceptional value. This distinct pricing advantage underscores the commercial viability of buffalo milk and its pivotal role in supporting the livelihoods and regional economies, distinguishing it from conventional milk sources. The premium price tag underscores the premium nutritional content and the potential for higher-quality dairy products derived from buffalo milk [3]. However, when it comes to average milk production, even the most productive buffalo breeds yield significantly less than Holstein cows, as evidenced by tests conducted on milk yields. This discrepancy highlights the substantial difference in milk-producing capabilities between the two species [4]. Indeed, increasing buffalo milk production while simultaneously improving its quality is a vital goal for fully realizing the economic potential of the buffalo milk industry. This objective, if achieved, would bring about significant benefits for the industry, making it an essential area of focus for researchers, farmers, and stakeholders.

Due to a variety of factors, including unstable estrus conditions and protracted calving intervals, traditional breeding methods for buffaloes pose significant challenges. These factors complicate the breeding process and necessitate innovative approaches to enhance the efficiency and success of buffalo reproduction [5]. Modern genetic technologies have introduced novel strategies for buffalo breeding, potentially simplifying the process. Among the various genetic variations, SNPs represent the most fundamental and prevalent form. Whole-genome sequencing is a powerful technique that enables the identification of genetic variations across the entire genome of an organism. This approach not only provides insights into the intrinsic genomic information but also plays a pivotal role in the study of human diseases, as well as in the breeding of crops and livestock [6]. Whole-genome association analysis serves as a potent tool for investigating complex genetic traits and pinpointing candidate genes. By scrutinizing genetic variations and polymorphisms throughout the entire genome, this method facilitates the identification of genomic regions and genes potentially associated with the traits of interest [7]. Globally, in previous years, several studies were conducted which analyzed the correlation between SNPs and milk production traits. In their study, Islam et al. [8] embarked on a comprehensive genome-wide association analysis involving 167 buffalo individuals, with the primary objective of discovering novel candidate genes linked to traits such as body weight and production performance. This analysis aimed to shed light on the genetic underpinnings of these economically important characteristics. The study employed the 90K Axiom Buffalo SNP Array to detect SNPs within a range of 2000 base pairs upstream of the buffalo *FABP* gene [9]. They identified SNP sites within a specific region and performed an association analysis to examine the relationship between these SNPs and various milk-related traits. Despite the increasing interest in the application of whole-genome association analysis in studying milk-related traits in water buffaloes, the number of studies conducted in this area remains limited when compared to the extensive body of research in cattle. This highlights the need for further investigation in this field to deepen our understanding of the genetic factors underlying milk production in water buffaloes.

The primary objective of this study was to identify key factors that influence lactation performance in water buffaloes, offering a new approach to enhancing breeding programs for improved milk production. By understanding these factors, we aim to provide valuable insights that will guide our future research and contribute to the development of more effective breeding strategies.

## 2. Materials and Methods

### 2.1. Ethics Statement

All finished work was conducted in accordance with national and international guidelines. The protocol for this study was approved by the Attitude of the Animal Care & Welfare Committee of the Guangxi Buffalo Research Institute (Approval Code: GXU2019-021).

### 2.2. Phenotypes and Animal Resources

The data for this study were collected from 120 water buffaloes, comprising 1 local Poyanghu water buffalo (DB), 46 hybrid water buffaloes (ZBs), 31 Murrah water buffaloes (MBs), and 42 Nili-Ravi water buffaloes (NBs), summing up to a total of 120. Among them, the hybrid water buffaloes are the offspring of more water buffaloes and local water buffaloes. These water buffaloes were born between the years 2000 and 2021. They were fed at the farm of Guangxi Buffalo Research Institute during the dry season from April to September. All records related to milk production are collected when all the water buffaloes are in their second calving. The initial test-day milk measurement is conducted from 5 to 70 days post-calving. The target traits of this study are 305-day milk yield (*MY*), peak milk yield (*PM*), total protein yield (*PY*), total milk fat yield (*FY*), fat percentage (*FP*), and protein percentage (*PP*). The calculation methods for *PP* and *FP* are as follows:$$FP=\frac{FY}{MY}$$$$\;PP=\frac{PY}{MY}$$

### 2.3. Sample Collection and Sequencing

Blood samples from water buffalo were obtained through tail vein puncture utilizing a vacuum blood collector. The genomic DNA was extracted from the blood using the phenol/chloroform method, and its integrity and yield were evaluated via agarose gel electrophoresis. The DNA libraries were sequenced on the Illumina sequencing platform (Illumina HiSeqTM 2000) by Genedenovo Biotechnology Co., Ltd. (Guangzhou, China).

### 2.4. Alignments and Variant Identification

The clean reads were aligned to the reference genome (UOA_WB_1) using BWA-MEM (v0.7.17) with default settings [10]. Then, Samtools (v1.9), Picard tools (v3.1.1), and GATK (v4.0) were used for SNP detection [11,12]. All detected SNPs underwent filtering through the “Variant Filtration” module of GATK, using the following standard parameters: variants with Quality Depth (QD) < 2; FS (Phred-scaled *p*-value using Fisher’s exact test for strand bias detection) > 60; MQRankSum (Z-score of the rank sum of the Phred-scaled mapping qualities) < −12.5; ReadPosRankSum (Z-score of the rank sum of the Phred-scaled position bias estimations) < −8; MQ (root mean square of the mapping quality) < 40.0; the mean sequencing depth of variants (across all individuals) was limited to less than 1/3× and more than 3×; SOR (strand odds ratio) > 3.0; the maximum missing rate was less than 0.1; and SNPs were limited to two alleles.

### 2.5. Variation Filtering

The presence of rare alleles (alleles with low frequency within the population), high rates of missing data, and substantial heterozygosity at specific loci can introduce anomalies in population analysis and whole-genome association studies. Therefore, we aligned the processed reads to the reference genome (UOA_WB_1). Subsequently, we employed the PLINK (v1.9) software to filter the detected loci based on standard criteria [13]. The filtering process involved stringent adherence to several criteria: exclusion of non-biallelic SNPs, removal of those with a minor allele frequency below 0.05, discarding SNPs with a missing genotype rate exceeding 20%, and further limiting the analysis to SNPs with a heterozygosity ratio below the threshold of 0.8. This was all executed using the robust PLINK (v1.9) software.

### 2.6. Principal Component Analysis

GCTA (v1.92.2) is a robust tool for the analysis of whole-genome complex traits [14]. In this study, we utilized the GCTA (v1.92.2) and PLINK (v1.9) software to perform PCA (principal component analysis) using the selected SNP markers. This analysis enabled us to derive the variance accounted for by each PC (principal component) and the score matrix representing the samples’ positions within each PC.

### 2.7. Population Structure Analysis

Population structure analysis offers valuable insights into the ancestry and composition of individuals, rendering it an exceptionally effective approach for elucidating genetic relationships. To validate the outcomes of PCA, we performed population structure analysis. Model-based population structure inference methods typically assume that the markers utilized for analysis are independent of one another. Consequently, prior to initiating the analysis, it is essential to execute marker independence filtering, which is based on the assessment of linkage disequilibrium between markers. In this analysis, we utilized the PLINK (v1.9) and Admixture software (v1.3) to perform marker filtering for population structure analysis. In our analysis, we implemented a 100 kb step size and a 10 nucleotide (nt) window size, and we removed one marker from each pair of markers with an r^2^ value greater than 0.2. Specifically, we removed the marker with the higher physical position from each pair of markers with a high degree of linkage disequilibrium. As a result of implementing the aforementioned filtering strategy, we retained a total of 99,261 markers for the population structure analysis. Utilizing the filtered SNP markers, we conducted a principal component analysis (PCA) using PLINK to investigate the population structure and clustering patterns. The PCA results were visualized to illustrate the relationships among the first three principal components, providing insights into the genetic diversity and relatedness within the population. Additionally, we employed the Admixture software (v1.3) to perform an in-depth analysis of population structure, estimating the proportion of ancestry from K ancestral populations and identifying subpopulations within the dataset. In our analysis, we explored the cross-validation (CV) error for various k-values using the Admixture software (v1.3), ranging from 2 to 9. We utilized the PopHelper software (v2.2.7) [15] to generate bar plots illustrating the genetic composition of each sample within every subgroup. By systematically testing these k-value hypotheses, we aimed to identify the optimal number of clusters that would provide the most meaningful and informative partitioning of the studied populations.

### 2.8. Genome-Wide Association Mapping

Our research focused on six primary dairy production traits: MY, PM, PY, FY, PP, and FP. By employing the TASSEL software (v5.2.54) [16], we executed the widely used General Linear Model (GLM) (Q) for genome-wide association studies. After Bonferroni correction, sites with *p*-values less than the given threshold 0.05/N (number of SNP) were selected as significant sites. SNPs with *p*-values below this threshold were considered highly significant and selected for further analysis. Subsequently, these SNPs were compared against the reference genome to pinpoint candidate genes for further investigation. It is anticipated that the identification of these genes will significantly contribute to a breeding program for buffaloes, leading to enhancements in both milk production quantity and the quality of buffalo dairy products.

Generalized Linear Models are a widely employed and versatile statistical method for data analysis [17]. In the present study, we utilized Generalized Linear Models for conducting a genome-wide association analysis. The GLM (*Q*) analysis model is expressed as follows:y=Xα+Qβ+cP+e

In this formula, *y* is the vector of phenotypes, *X* is the genotype matrix, α is the vector of genotype effects, P is the PCA variance_explained matrix, c is the vector of PCA variance effects, e is the vector of residual effects, and *Q* refers to the fixed-effect matrix, which represents calf gender, calving year, and herds. The outcomes are presented through Manhattan plots and Q-Q plots. SNPs exhibiting *p*-values below the specified threshold 0.05/N (number of SNP) are identified as highly significant SNPs. When a reference genome is available, candidate genes are determined by including those genes that are physically positioned within a 50 kb genomic region surrounding the significant SNPs. DbSNP [18] is a database specifically designed by NCBI to store genetic variation information. We use dbSNP to determine whether the SNPs we have identified are located in the coding regions of genes.

### 2.9. Pathway Enrichment and Protein–Protein Interaction

Genes often work in concert to perform specific biological functions. Pathway-based analysis is a valuable approach for understanding the roles of genes in these complex processes. The KEGG (Kyoto Encyclopedia of Genes and Genomes) database [19] stands as one of the foremost publicly accessible resources for pathway-related data. To identify significantly enriched metabolic and signal transduction pathways among CAGs (Candidate-Associated Genes) relative to the entire genome context, pathway enrichment analysis was performed. The method for calculating enrichment is consistent with that employed in Gene Ontology (GO) [20] analysis:P=1−∑i=0m−1(Mi)(N−Mn−i)(Nn) 

In this context, *N* signifies the total count of genes with KEGG annotations, while *n* denotes the number of CAGs within *N*. *M* represents the total number of genes annotated to particular pathways, and *m* is the number of CAGs in *M*. Following the calculation of the *p*-value, it was corrected using False Discovery Rate (FDR) adjustment, with an FDR value of 0.05 or less being set as the threshold. Pathways that meet this criterion are categorized as significantly enriched pathways in CAGs. Finally, we utilize the String database to identify genes that are significantly represented in pathways and create a protein–protein interaction map. Gene Ontology (GO) and Kyoto Encyclopedia of Genes and Genomes (KEGG) analyses were performed using the OmicShare tools, a free online platform for data analysis (http://www.omicshare.com/tools, accessed on 9 January 2025).

### 2.10. Statistical Analysis

In the analysis of trait correlations, we used Pearson’s correlation coefficient to quantify the linear relationships between different traits. The Pearson’s correlation coefficient ranges from −1 to 1, where values close to 1 indicate a strong positive correlation, values close to −1 indicate a strong negative correlation, and values around 0 indicate no significant correlation. Statistical analyses were performed using the SPSS 18.0 software package (SPSS Science, Chicago, IL, USA). Experimental data were subjected to t-test and ANOVA analyses, with a significance threshold set at *p* < 0.05. Graphs were generated using GraphPad Prism 8 software (GraphPad, Santiago, MN, USA). Data are presented as mean ± standard deviation (SD).

## 3. Results

### 3.1. Phenotypic Value Statistics of the Traits

During the phenotypic evaluation of buffalo. We carried out MY, PM, PY, PP, FP, and FY phenotypic value analysis.

For milk production traits, the mean value of MY was recorded as 2321.3 kg, while the mean value of PM was measured as 11.4 kg. The mean value of FY was recorded as 116.1 kg, while the mean value of PY was measured as 80.8 kg. The average protein and fat percentages in our population were 4.8% and 5.2%, respectively. The phenotypic data statistics are presented in Table 1.

Through the correlation analysis of various lactation traits, it was found that the correlation between total protein yield (PY) and total milk fat yield (FY) is the highest, reaching above 0.9 (Figure 1). This strong positive correlation suggests a significant relationship between the production of protein and fat in milk, indicating that these two components tend to vary together in response to genetic or environmental factors.

In this study, a comprehensive analysis of genome-wide variations led to the identification of 2,208,174 genetic markers. Among the detected genetic markers, 2,012,270 were identified as SNPs and 195,904 were classified as insertion–deletion (Indel) variants. Following stringent filtering criteria, a refined set of 99,261 markers was retained, comprising 93,494 SNPs and 5767 Indels.

### 3.2. Population Structure

Upon obtaining PCA scores, the samples under investigation can be visualized via a scatter plot that utilizes the values of the first three principal components as axes. Referring to Figure 2, in scatter plots Figure 2A, it is evident that the majority of individuals within herds MB and NB are distinctly isolated from one another. In the scatter plot in Figure 2B, we can observe clustering, particularly in groups MB and ZB, where these two clusters overlap and are closely grouped together in multiple instances. In Figure 2A,C, it is evident that the herds are broadly segregated into three distinct clusters. One cluster predominantly consists of NB herds, whereas the other two clusters are primarily composed of MB, ZB, and DB herds, respectively.

To establish the optimal number of clusters (k), we used the Admixture software and evaluated cross-validation error rates. The Admixture algorithm performs model-based clustering and estimates the proportion of ancestry from K ancestral populations. By minimizing the cross-validation error rates, we identified the value of k = 3 that best fits our data. Figure 3 displays the line graph depicting the cross-validation error rate.

To simulate the population classification and genetic ancestry of each sample across varying numbers of subgroups (K = 2–9), we utilized the PopHelper software (v2.2.7) [15] to generate bar plots illustrating the genetic composition of each sample within every subgroup. The results are presented in Figure 4, where each color corresponds to a distinct cluster for each K-value. From the line graph of the cross-validation error rate, it is evident that the optimal number of clusters is K = 3. Similarly, as observed in the bar graph (Figure 3) depicting the genetic composition of the samples, when K = 3, it is the optimal number of clusters for these 120 buffaloes. This finding is consistent with the results obtained from the PCA. Consequently, we conclude that these 120 buffaloes can be effectively divided into three distinct subgroups.

### 3.3. Results of the Genome-Wide Associations

Following the calculation of *p*-values for the SNP loci using a Generalized Linear Model, we constructed a Manhattan plot and a Q-Q plot, as presented in Figure 5.

The leftmost plot is a Manhattan plot, where the visually discernible blue line, running parallel to the *x*-axis, serves as the critical demarcation line. In the Manhattan plot, the points that rise above the threshold line, which is the blue line paralleling the *x*-axis, signify significant loci. Upon identifying the significant loci that surpass the threshold line in the Manhattan plot, the subsequent step involves documenting the relevant information pertaining to these significant loci.

The plot on the right is the Q-Q plot, where the points in the bottom left corner fall along the line, suggesting that the observed *p*-values align closely with the expected values. The points exhibit a distinct upward deviation from the diagonal in the upper right corner, which signifies that the observed *p*-values exceed the anticipated values. The presence of these points, which denote significant loci, across all four plots underscores the validity of the analytical model, suggesting its appropriateness in capturing the underlying patterns. Following this, the genes situated within a 50 kb range of the significant loci are carefully selected and earmarked as candidate genes.

In our GWAS, we have identified a large number of SNPs associated with FY and PY (Figure 5). We identified 56 statistically significant SNPs and 54 candidate genes within a 50 Kb range surrounding these loci that were associated with the traits MY, PY, PP, FP, and FY (Table 2).

### 3.4. Kyoto Encyclopedia of Genes and Genomes Pathway Analysis of Candidate Genes

GO and KEGG analyses were conducted to examine the functional pathways for DEGs. The top 20 GO terms, classified by –log10(*p*-value), were significantly enriched in candidate genes compared to the genome background GO. The top 20 GO terms included eight biological processes (BPs), a cellular component (CC) and one molecular function (MF). The BP terms were enriched in growth (GO:004007), developmental processes (GO:0032502), cellular processes (GO:0009987), metabolic processes (GO:0008152), and cellular component organization or biogenesis (GO:0071840) (Figure 6A, Appendix A).

The functional enrichment cycle diagram displays the top 20 KEGG pathways (Figure 6B, Appendix A), classified by –log10 (*p*-value), which reveals that the candidate genes were mainly enriched in five KEGG_A_class pathways, including metabolism, environmental information processing, cellular processes, organismal systems, and human diseases. Among these pathways, environmental information processing included ABC transporters (ko02010), the cAMP signaling pathway (ko04024), the TGF-β signaling pathway (ko04350), and the wnt signaling pathway (ko04310); metabolism included arginine and proline metabolism (ko00330) and phenylalanine metabolism (ko00360); biosynthesis of amino acids (ko01230) and metabolic pathways was involved (ko01100); and cellular processes included signaling pathways regulating pluripotency of stem cells (ko04550). In conclusion, our data suggest that the identified candidate genes play a crucial role in regulating lactation performance, particularly in terms of milk fat yield (FY), protein percentage(PP), fat percentage (FP), and protein yield (PY), by modulating the tgf-β signaling pathway, wnt signaling pathway, metabolic pathways, and cAMP signaling pathway. These findings provide valuable insights into the molecular mechanisms underlying dairy production and could inform future breeding strategies to enhance milk quality and quantity.

### 3.5. Significant Association of Milk Protein Content with SNP Validation

Table 3 presents the results of individual genotyping for four key loci in water buffalo: NC_037557.1:18868198 (*ABCC4*), NC_037549.1:52254091 (*LEFTY2*), NC_037559.1:81684074 (*DGAT1*), and NC_037551.1:87385802 (*CAMK2D*). The analysis reveals significant differences in milk protein content across different genotypes at these loci.

For *ABCC4* (NC_037557.1:18868198), the A/A genotype is associated with significantly lower milk protein yield compared to the G/A and G/G genotypes.

For *LEFTY2* (NC_037549.1:52254091), both the A/A and G/A genotypes exhibit significantly lower milk protein yield compared to the G/G genotype.

For *DGAT1* (NC_037559.1:81684074), the T/T and C/T genotypes are associated with significantly lower milk protein yield compared to the C/C genotype.

For *CAMK2D* (NC_037551.1:87385802), the T/T genotype shows significantly lower milk protein yield compared to both the CT and C/C genotypes.

These findings highlight the influence of specific genotypes on milk protein yield, providing valuable insights for genetic selection and breeding programs aimed at improving milk quality in water buffalo.

## 4. Discussion

### 4.1. Population Stratification

Population stratification is a pivotal factor in GWASs, as it significantly influences the findings due to disparities in ancestral origins. These differences can give rise to discrepancies in allele frequencies across populations, potentially generating spurious association signals [21,22]. To mitigate the risk of false-positive findings in our analysis, it is imperative to acknowledge the existence of population stratification. PCA is capable of diminishing the complexity of a dataset while keeping its covariance structure intact [23]. To assess the classification effectiveness of the first 10 principal components, we calculated the proportion of variance explained by each component using the filtered SNP markers with PLINK. This analysis helped us understand how well the principal components capture the genetic variation within the population. Furthermore, we used the Admixture software to perform K-cluster analysis and evaluated the cross-validation error (CV-error) to determine the optimal number of subpopulations. By minimizing the CV-error, we identified the most appropriate value of K for population stratification. Though these buffaloes are raised on the same farm, their origins differ: the Nili-Ravi buffalo hails from Pakistan, while the Murrah buffalo originates from India. From our PCA plot, we observe clustering, particularly between groups MB and ZB, where these two clusters overlap and coalesce in several instances. This indicates that the buffaloes in these two groups may have a closer genetic relationship compared to the other groups. Overall, these groups can be generally categorized into two distinct segments. The plot depicting cross-validation error suggests that the optimal value of K, which corresponds to the lowest cross-validation error, is 3. Therefore, it is concluded that these 120 water buffaloes should be divided into three subgroups.

The simple linear model serves as a valuable instrument for conducting SNP and phenotype analysis, concurrently managing population stratification by incorporating the relevant population structure as a covariate [24]. Typically, the models used in GWAS analysis are adjusted based on the genetic background and stratification of the dataset. In general, after correcting for population stratification, the inflation factor should approach a value of 1 in instances that adhere to a normal distribution [24]. In our experimental outcomes, the inflation factor for MY was 1.056, that for PM was 1.1, that for PY was 0909, that for PP was 1.06, that for FP was 1.12, and that for total FY was 0.916. These findings suggest that our model’s adjustment for population stratification is valid.

### 4.2. Genome-Wide Association Analysis of Milk Production-Related Traits

Enhancing milk production and quality in water buffalo has emerged as a critical research priority within the dairy industry. Water buffalo play a vital role in global dairy production, particularly in regions where they are the primary milk source. However, there is a significant research gap in understanding the lactation traits of water buffalo. Unlike Holstein cows, which have been extensively studied through detailed production records and numerous genome-wide association studies (GWASs), water buffalo have received considerably less attention in these areas. This disparity in research means that key aspects of water buffalo genetics, such as the identification of candidate genes and pathways influencing milk yield, protein content, and fat composition, remain underexplored. As a result, breeding programs for water buffalo are not as advanced or effective as those for other dairy species, limiting the potential for improving milk production and quality. Addressing this research gap is essential for developing targeted breeding strategies and enhancing the overall productivity of water buffalo dairy operations.

For milk production traits, this investigation assessed and quantified the MY as well as the PM. The mean values were 2321.3 kg and 11.4 kg, respectively. This investigation also assessed and quantified the PY, PP, FP, and FY. The mean values were 80.8 kg, 4.8%, 5.2%, and 116.1 kg, respectively. The mean milk yield (MY) of our buffalo population is lower than that of Mediterranean buffaloes (2321.3 kg) but higher than that of Brazilian buffaloes (1578.90 kg). In contrast, the mean values for peak milk yield (PM), fat yield (FY), and protein yield (PY) are more closely aligned with those of Mediterranean buffaloes [25,26]. Therefore, we are confident that our production record measurements are within the normal range and can be converted into corresponding phenotype vectors in the model. We carried out an association analysis for each of the phenotypic traits.

In the association analysis of PY and FY, we identified a total of 54 candidate genes. Lactation is a highly complex process involving the coordinated action of multiple signaling pathways. Among the identified genes, several are associated with key pathways, including the cAMP signaling pathway (*ABCC4*), TGF-β signaling pathway (*LEFTY2*), Wnt signaling pathway (*CAMK2D*), and metabolic pathways (*DGAT1*).

The *ABCC4* gene, located on chromosome NC_037557.1, encodes ATP-binding cassette (ABC) transporters. These proteins are essential for modulating platelet aggregation, a critical process in blood clotting and hemostasis [27]. Moreover, *ABCC4* has been detected in milk from the early stages of lactation [28]. Our findings indicate that *ABCC4* may regulate milk production by influencing the cAMP signaling pathway, suggesting its potential importance in lactation biology.

The TGF-β signaling pathway (*LEFTY2*) plays a pivotal role in mammary gland development and lactation [29,30,31]. Studies have shown that the TGF-β pathway not only regulates the proliferation and differentiation of mammary epithelial cells but also influences the synthesis and secretion of milk components. Specifically, TGF-β promotes the self-renewal of mammary stem cells while inhibiting excessive cell proliferation, ensuring the proper development and function of mammary tissue.

The Wnt signaling pathway is a complex regulatory network that plays a crucial role in controlling cell growth, differentiation, and tissue development. In particular, breast development is intricately linked to lactation performance, as the proper formation and function of the mammary gland are essential for efficient milk production. Extensive previous research has shown that the gene expression and epigenetic regulation of *CAMK2D* (Calcium/Calmodulin-Dependent Protein Kinase II Delta) significantly impact the physiological functions of the mammary gland. *CAMK2D* is involved in various cellular processes, including calcium signaling and gene transcription, which are critical for mammary gland development and lactation [32,33].

Milk fat provides essential fatty acids (FAs) that can contribute to various health benefits, such as supporting cardiovascular health, enhancing nutrient absorption, and promoting overall well-being, depending on their specific composition. Previous research has demonstrated that polymorphisms in the *DGAT1* (Diacylglycerol O-Acyltransferase 1) gene influence various milk production traits, including milk yield, fat yield, protein yield, and the fat and protein content of milk [34,35,36]. Consistent with these findings, our study revealed that *DGAT1* polymorphisms have a significant impact on fat content, further highlighting the importance of this gene in determining milk composition.

### 4.3. The Mechanism of SNP Mutation and the Milk Production Traits

Among the selected SNPs, none are non-synonymous. However, we identified base mutations in non-coding regions, specifically at positions NC_037557.1:18868198, NC_037549.1:52254091, NC_037559.1:81684074, and NC_037551.1:87385802, which may potentially influence milk production-related traits. These results suggest that variation in non-coding regions may play an important role in the regulation of these important production and quality traits. The variability of non-transcriptional regulatory sequences (e.g., promoters, enhancers, CTCF sites) is closely related to the mechanism of non-coding variation in cell development, but whether this rule applies to milk production traits needs further verification [37,38].

## 5. Conclusions

In this investigation, genome-wide association analysis was performed on four milk production-related characteristics in domestic water buffaloes, which included MY, PM, PY, and FY. A total of 56 significant SNP loci were identified, and 54 candidate genes within a 50 Kb range surrounding these loci were selected. These candidate genes were enriched in biological processes such as the cAMP signaling pathway (*ABCC4*), the TGF-β signaling pathway (*LEFTY2*), the Wnt signaling pathway (*CAMK2D*), and metabolic pathways (*DGAT1*), all of which are directly or indirectly involved in the lactation process. These findings offer a reference framework for comprehending the genetic architecture underlying milk production and quality attributes in water buffaloes, thereby paving the way for subsequent biological validation of the implicated genes. This is crucial guidance for breeding and improvement programs in water buffaloes. In future studies, we will focus on elucidating the relationships between candidate genes and metabolites involved in the lactation process to uncover the metabolic pathways and mechanisms through which these genes influence milk production and composition.

## Figures and Tables

**Figure 1 genes-16-00163-f001:**
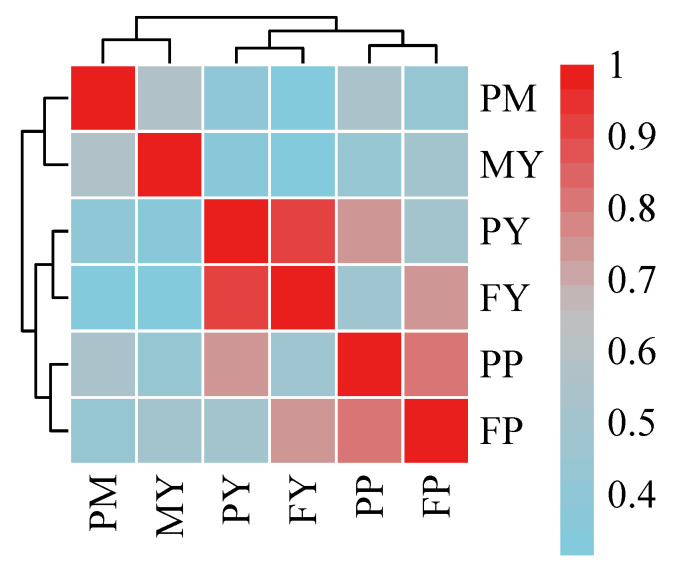
Correlation analysis of various lactation traits. MY, milk yield; PM, peak milk yield; PY, protein yield; FY, fat yield; PP, protein percentage; FP, fat percentage.

**Figure 2 genes-16-00163-f002:**
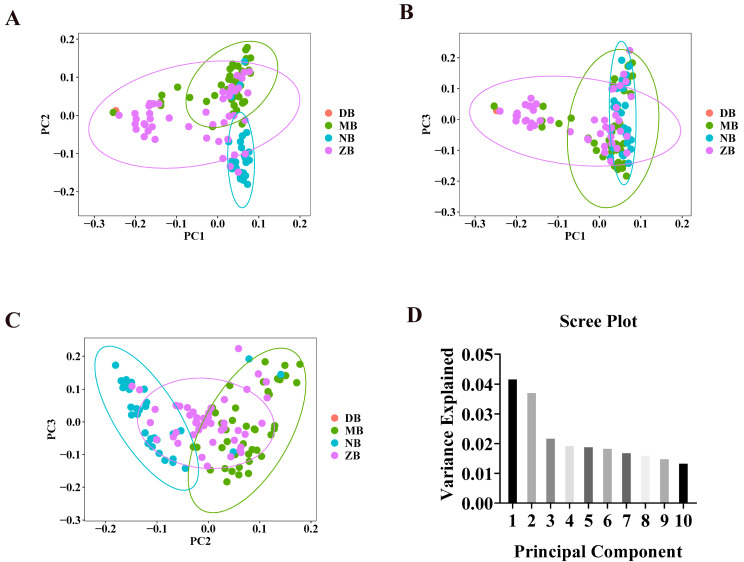
The sample clustering obtained from PCA through three two-dimensional scatter plots, namely scatter (**A**), scatter (**B**), and scatter (**C**); scree plot (**D**). The percentage of variance explained by each PC is noted in parentheses. In the scatter plots, colored circles represent four different groups: DB, MB, NB, and ZB correspond to 1 DB, 42 MBs, 31 NBs, and 46 ZBs, respectively.

**Figure 3 genes-16-00163-f003:**
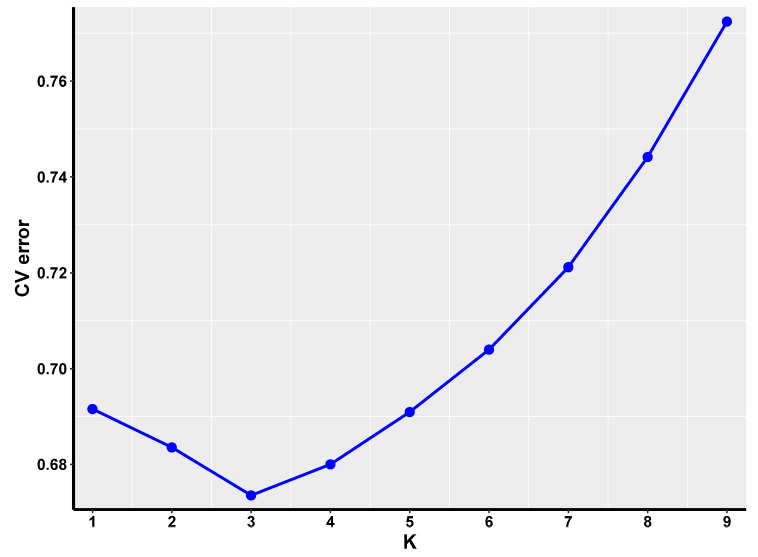
The line graph illustrating the cross-validation error rate is depicted, with the number of sample clusters delineated along the *x*-axis and the corresponding cross-validation error rate indicated on the *y*-axis.

**Figure 4 genes-16-00163-f004:**
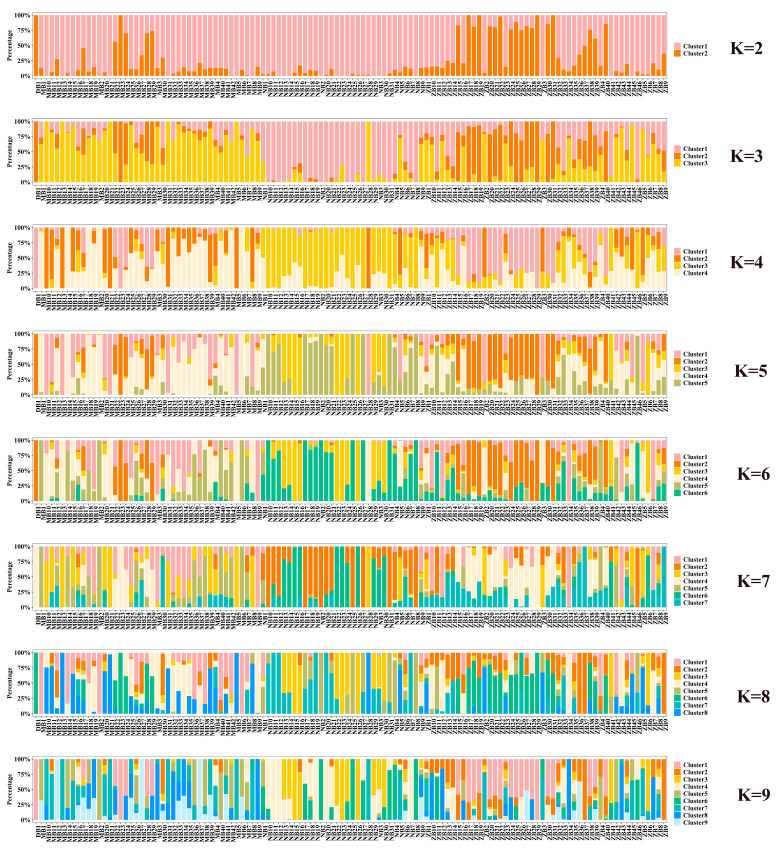
Genetic bar chart illustration for K-means clustering with varying numbers of clusters (K = 2 to 9).

**Figure 5 genes-16-00163-f005:**
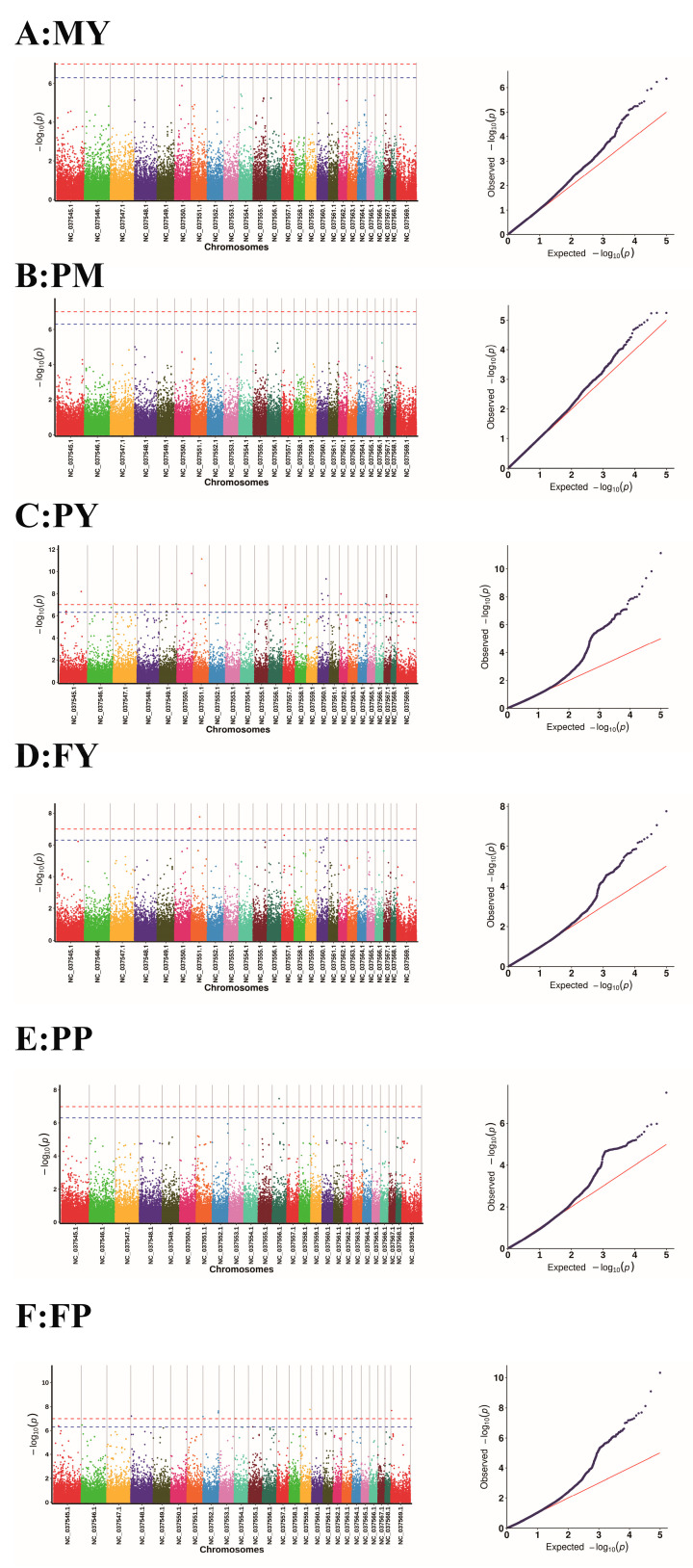
Association analysis with milk production-related traits in water buffalo was conducted using the GLM-Q approach. The traits investigated include MY (**A**), PM (**B**), PY (**C**), FY (**D**), PP (**E**), and FP (**F**). The Manhattan plot on the left, created using the qqman package, illustrates the *p*-values for SNP markers across 25 chromosomes (comprising 24 autosomes and 1 X chromosome). The blue line delineating the Manhattan plot signifies the significance threshold, determined by 0.05/N (number of SNP). Markers that surpass this threshold are deemed significant. The plot on the right is a Q-Q plot, where the *x*-axis denotes the observed values of the markers, and the *y*-axis represents the expected values, which have been transformed into the −10 log scale.

**Figure 6 genes-16-00163-f006:**
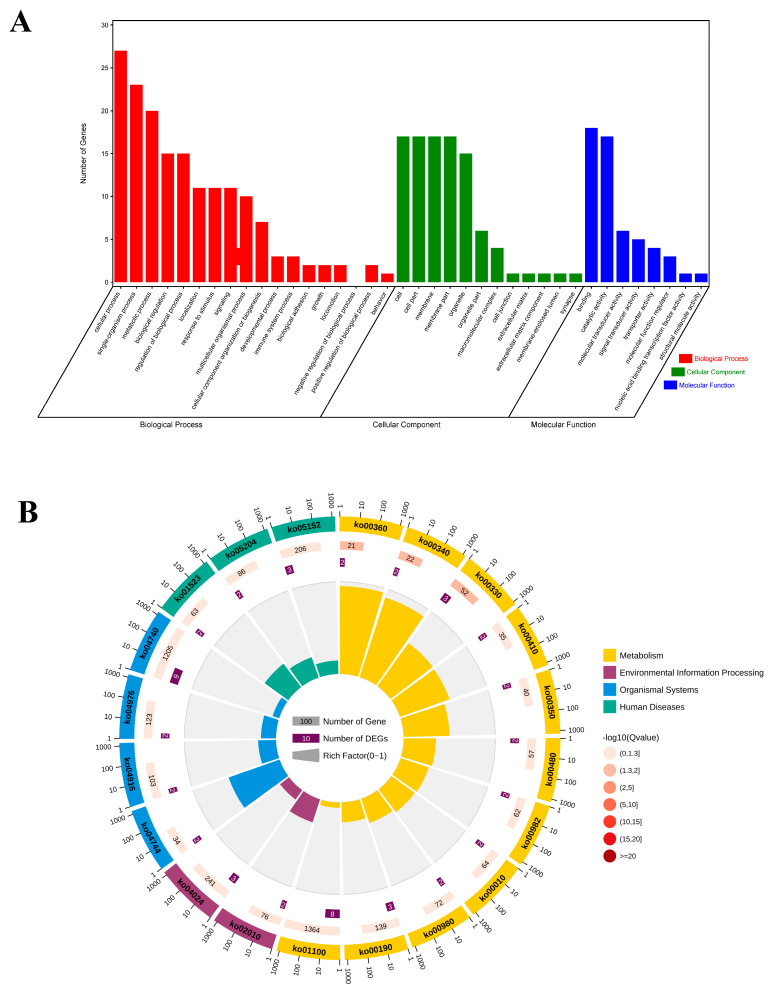
GO and KEGG analysis of candidate genes. (**A**) GO bar plot diagram showing the top 20 enriched GO terms. GO categories, including cellular component, biological process, and molecular function. (**B**) The enrichment circle diagram shows the KEGG analysis of the top 20 pathways. Four circles from the outside to the inside. First circle: the classification of enrichment; outside the circle is the scale of the number of genes. Different colors represent different categories. Second circle: number and *p*-values of the classification in the background genes. The more genes, the longer the bars; the smaller the value, the redder the color. Third circle: bar chart of the total number of candidate genes. Fourth circle: rich factor value of each classification (number of candidate genes in this classification divided by the number of background genes). Each cell of the background helper line represents 0.1, and the color coding signifies the statistical significance of the corresponding enrichment.

**Table 1 genes-16-00163-t001:** Statistical description of lactation traits *.

Traits	Mean	SD	Min	Max
MY	2321.3	861.8	480.8	5185.3
PM	11.4	4.8	2.7	24.4
PY	80.8	41.2	4.0	275.8
FY	116.1	57.9	5.0	363.3
PP	4.8	0.4	4.4	5.8
FP	5.2	0.8	3.7	8.9

* MY, milk yield; PM, peak milk yield; PY, protein yield; FY, fat yield; PP, protein percentage; FP, fat percentage; SD, standard deviation.

**Table 2 genes-16-00163-t002:** The SNPs identified via genome-wide association analysis encompass detailed information regarding their chromosomal locations, *p*-values, and associated candidate genes.

Traits	SNP	Chr	Pos	*p*	R^2^	Candidate Genes
MY	1	NC_037552.1	110795896	4.3 × 10^−7^	0.39	*CNTNAP2*
PY	2	NC_037545.1	45287655	4.01 × 10^−7^	0.36	--
PY	3	NC_037545.1	45287667	4.01 × 10^−7^	0.36	--
PY	4	NC_037545.1	45287677	4.01 × 10^−7^	0.36	--
PY	5	NC_037545.1	45287689	4.01 × 10^−7^	0.36	--
PY	6	NC_037545.1	155842848	6.41 × 10^−7^	0.43	*KCNAB1*
PY	7	NC_037546.1	174939089	1.82 × 10^−7^	0.37	*TINAGL1*; *AZIN2*
PY	8	NC_037547.1	10261615	8.43 × 10^−8^	0.35	*RBFOX3*
PY	9	NC_037548.1	58828968	3.56 × 10^−7^	0.33	*NEDD1*
PY	10	NC_037548.1	97643871	9.64 × 10^−8^	0.369	*EEA1*; *PLEKHG7*
PY	11	NC_037549.1	52254091	4.08 × 10^−7^	0.31	*LEFTY2*; *PYCR2*; *-*
PY	12	NC_037549.1	122104312	8.972 × 10^−8^	0.41	*DOC2G*; *NUDT8*; *TBX10*; *ALDH3B1*; *UNC93B1*; *ALDH3B1*; *NDUFS8*; *TCIRG1*
PY	13	NC_037550.1	7990287	2.51 × 10^−7^	0.34	*CFAP126*; *SDHC*
PY	14	NC_037550.1	109178670	1.5 × 10^−9^	0.48	*MAP7D1*; *TRAPPC3*; *COL8A*; *ADPRHL2*; *TEKT2*
PY	15	NC_037551.1	62424439	7.20 × 10^−11^	0.54	--
PY	16	NC_037551.1	87385802	1.7 × 10^−9^	0.43	*CAMK2D*
PY	17	NC_037552.1	119759448	4.64 × 10^−7^	0.30	*USP17L13*
PY	18	NC_037556.1	9792358	3.18 × 10^−7^	0.36	--
PY	19	NC_037557.1	18868198	1.71 × 10^−7^	0.34	*ABCC4*
PY	20	NC_037557.1	18868200	1.72 × 10^−7^	0.34	*ABCC4*
PY	21	NC_037557.1	18868226	1.71 × 10^−7^	0.34	*ABCC4*
PY	22	NC_037557.1	18868442	2.02 × 10^−7^	0.34	*ABCC4*
PY	23	NC_037557.1	18868443	2.02 × 10^−7^	0.34	*ABCC4*
PY	24	NC_037557.1	18868449	2.02 × 10^−7^	0.34	*ABCC4*
PY	25	NC_037557.1	18868500	1.71 × 10^−7^	0.34	*ABCC4*
PY	26	NC_037557.1	18868507	1.71 × 10^−7^	0.34	*ABCC4*
PY	27	NC_037557.1	18868523	1.71 × 10^−7^	0.34	*ABCC4*
PY	28	NC_037560.1	28753781	9.90 × 10^−8^	0.40	*GUCY2D*; *LRRC32*
PY	29	NC_037560.1	35600620	3.36 × 10^−8^	0.40	*OR52Z1*; *OR51V1*; *OR51V1*; *OR52A5*; *OR52K1*; *OR52K1*
PY	30	NC_037560.1	60745000	4.5 × 10^−10^	0.44	*TMPRSS5*
PY	31	NC_037560.1	74611905	1.48 × 10^−8^	0.38	*CNTN5*
PY	32	NC_037562.1	12469915	1.04 × 10^−8^	0.43	--
PY	33	NC_037564.1	56654985	8.12 × 10^−8^	0.39	--
PY	34	NC_037564.1	56655021	8.12 × 10^−8^	0.39	--
PY	35	NC_037565.1	13765148	1.19 × 10^−7^	0.37	*CTNNB1*
PY	36	NC_037566.1	61557795	1.04 × 10^−7^	0.37	*KCNG2*; *PQLC1*; *TXNL4A*; *YVCT*
PY	37	NC_037567.1	17641416	1.31 × 10^−8^	0.39	*TLL2*; *TM9SF3*
PY	38	NC_037567.1	17641470	1.91 × 10^−8^	0.39	*TLL2*; *TM9SF3*
PY	39	NC_037567.1	17641671	1.31 × 10^−8^	0.39	*TLL2*; *TM9SF3*
PY	40	NC_037567.1	44281645	8.32 × 10^−8^	0.34	--
FY	41	NC_037550.1	109178670	8.94 × 10^−8^	0.37	*MAP7D1*; *TRAPPC3*; *COL8A*; *ADPRHL2*; *TEKT2*
FY	42	NC_037551.1	62424439	1.7 × 10^−8^	0.42	--
FY	43	NC_037557.1	18651732	2.48 × 10^−7^	0.32	*ABCC4*
FY	44	NC_037560.1	60745000	4.44 × 10^−7^	0.32	*TMPRSS5*
FY	45	NC_037560.1	74050792	3.67 × 10^−7^	0.36	--
PP	46	NC_037556.1	50669172	3.34 × 10^−8^	0.43	--
FP	47	NC_037545.1	35877328	4.11 × 10^−7^	0.30	--
FP	48	NC_037546.1	6705158	3.44 × 10^−7^	0.27	--
FP	49	NC_037548.1	4405934	6.25 × 10^−8^	0.33	*FAM118A*; *UPK3A*; *KIAA0930*
FP	50	NC_037552.1	24460	6.65 × 10^−8^	0.29	--
FP	51	NC_037552.1	112491377	3.27 × 10^−8^	0.32	*ZNF777*; *ZNF746*
FP	52	NC_037552.1	113607118	2.36 × 10^−8^	0.33	*ABCB8*; *ASIC3*
FP	53	NC_037555.1	29774524	2.45 × 10^−7^	0.28	*PRKCH*
FP	54	NC_037559.1	81684074	5.71 × 10^−8^	0.32	*DGAT1*; *HSF1*
FP	55	NC_037564.1	36725181	9.35 × 10^−8^	0.31	*LINGO1*
FP	56	NC_037569.1	5832499	2.09 × 10^−8^	0.35	*KAL1*

**Table 3 genes-16-00163-t003:** The results of individual genotyping *.

Candidate Genes	SNP (Chr:Pos)	Milk Protein Yield		
		Homozygous Mutation	Heterozygous Mutation	Reference Genotype
*ABCC4*	NC_037557.1:18868198	A/A	G/A	G/G
		73.46 ± 29.68C	103.97 ± 47.99B	144.57 ± 65.04A
*LEFTY2*	NC_037549.1:52254091	A/A	G/A	G/G
		73.87 ± 31.82B	97.07 ± 19.41B	151.03 ± 74.08A
*DGAT1*	NC_037559.1:81684074	T/T	C/T	C/C
		67.01 ± 38.53B	90.62 ± 45.36B	125.61 ± 71.25A
*CAMK2D*	NC_037551.1:87385802	C/C	C/T	T/T
		62.03 ± 48.35C	95.85 ± 58.65B	133.24 ± 79.52A

* The phenotypic values of milk protein content are expressed as “least squares mean ± standard deviation”. Different letters in the same column indicate significant differences (*p* < 0.05); the same letter or no letter indicates no significant difference (*p* > 0.05). The genotypes of SNP loci are arranged in the order of homozygous mutant, heterozygous, and reference types.

## Data Availability

The data presented in this study are openly available in The European Variation Archive at https://www.ebi.ac.uk/eva/?eva-study=PRJEB72029, accessed on 15 September 2024.

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
