# Peer review of "Genome-Wide Association Studies for Lactation Performance in Buffaloes"

_genes, 2025, doi:10.3390/genes16020163_

Round 1

Reviewer 1 Report

Comments and Suggestions for Authors

Manuscript ID genes-3427964

In the manuscript entitled: 'Genome-Wide Association Studies for Lactation Performance in Buffaloes', the authors performed an integrated analysis of genomic sequencing data from 120 water buffaloes (4 breeds), the high-quality water buffalo genome assembly designated UOA_WB_1, and milk production traits including 305-day milk yield (MY), peak milk yield (PM), total protein yield (PY) and total milk fat yield (FY). The authors identified 31 significant SNPs and, based on these markers, selected 25 candidate genes enriched in lactation-related pathways, such as the cAMP pathway (ABCC4), the TGF-beta pathway (LEFTY2), the cGMP-PKG pathway (CNGA1) and the phosphatidylinositol pathway (PLCD3). These candidate genes (e.g. ABCC4, LEFTY2, CNGA1, PLCD3) provide a substantial theoretical basis for molecular breeding to improve milk yield and milk quality traits in buffaloes.

The manuscript is interesting and in line with the aims of the journal. Here are my observations, point by point:

Line 30: water; rewrite as Water

Line 76: This rigorous; delete the word “rigorous” from the whole text.

Lines 85: Describe the aim of the study clearly and unambiguously. The purpose of this study was ....

Line 92-94: 2.2. Phenotypes and animal resources

The data for this study was collected from 120 water buffaloes, comprising 1 local water buffalo (DB), 46 hybrid water buffaloes (ZB), 31 Murrah water buffaloes (MB), and 42 Nili-Ravi water buffaloes (NB)

I suggest to the authors to give the full name of the DB breed, (Dechuang, Dehong, Diangdongnan .... other?),

Lines 151: aforementioned pruning; The term 'pruning' is not correct, check and re-write it throughout the text.

Lines 160-161: 2.8. Genome-wide association mapping

Our research focused on six primary dairy production traits: MY, PM, PY and FY. I only read four traits, not six. Check and rewrite

Lines 174-176: In this formula, y is the vector of phenotypes, X is the genotype matrix, α is the vector of genotype effects, and Q refers to the fixed effects matrix, which represents represents calf gender, calving year and herds.

Is the error "e"?

Lines 218-220: Through the correlation analysis of various lactation traits, it was found that the correlation between total protein yield (PY) and total milk fat yield (FY) is the highest reaching above 0.9 (Figure 1).

0.9 is Pearson's coefficient? Specify in "statistical analysis".

Lines 399-438: 4.2. Genome-Wide Association analysis of milk production related traits.

My suggestion to the authors is a more clear-cut rewrite of this section.

Lines 443-444: These findings suggest that variations in non-coding regions could play a significant role in regulating these important agricultural traits.

“agricultural” is not correct term, maybe “These results suggest that variation in non-coding regions may play an important role in the regulation of these important production and quality traits”.

Lines 482: References. All references must be double-checked.

Lin, Y.X.; Sun, H.; Shaukat, A.; Deng, T.X.; Abdel-Shafy, H.; Che, Z.X.; Zhou, Y.; Hu, C.M.; Li, H.Z.; Wu, Q.P.; et al. Novel Insight Into the Role of ACSL1 Gene in Milk Production Traits in Buffalo. Front Genet 2022, 13, doi:ARTN 89691010.3389/fgene.2022.896910.

Correct form is

 Lin, Y.X.; Sun, H.; Shaukat, A.; Deng, T.X.; Abdel-Shafy, H.; Che, Z.X.; Zhou, Y.; Hu, C.M.; Li, H.Z.; Wu, Q.P.; Yang, L.; and Hua, G. Novel Insight Into the Role of ACSL1 Gene in Milk Production Traits in Buffalo. Front Genet 2022, 13, doi:ARTN 89691010.3389/fgene.2022.896910.

Pisanu, S.; Cacciotto, C.; Pagnozzi, D.; Puggioni, G.M.G.; Uzzau, S.; Ciaramella, P.; Guccione, J.; Penati, M.; Pollera, C.; Moroni, P.; et al. Proteomic changes in the milk of water buffaloes (water buffalo) with subclinical mastitis due to intramammary infection by Staphylococcus aureus and by non-aureus staphylococci. Sci Rep 2019, 9, 15850, doi:10.1038/s41598-019-52063-2.

Correct form is

Pisanu, S.; Cacciotto, C.; Pagnozzi, D.; Puggioni, G.M.G.; Uzzau, S.; Ciaramella, P.; Guccione, J.; Penati, M.; Pollera, C.; Moroni, P.; Bronzo, V.; Addis, M.F. Proteomic changes in the milk of water buffaloes (water buffalo) with subclinical mastitis due to intramammary infection by Staphylococcus aureus and by non-aureus staphylococci. Sci Rep 2019, 9, 15850, doi:10.1038/s41598-019-52063-2.

Author Response

Thank you very much for your valuable feedback. Your suggestions have greatly improved the research prospects and overall quality of our manuscript. We have carefully addressed each of your comments and have made the necessary revisions. Our detailed responses to your feedback are provided in the attached document. We appreciate your attention to this matter.

Reviewer 2 Report

Comments and Suggestions for Authors

The article is original and very relevant for the field. The authors realized an integrated analysis of genomic sequencing data from 120 water buffalo, the high-quality water buffalo genome assembly designated as UOA_WB_1, and milk production traits, including 305-day milk yield (MY), peak milk yield (PM), total protein yield (PY), and total milk fat yield (FY).

The results showed that authors identified 31 significant SNPs, and based on these markers, 25 candidate genes were selected. These findings may open new research directions, providing a substantial theoretical foundation for molecular breeding to enhance milk production in buffaloes and promoting the biodiversity.

The methodology of the study is modern and complex, authors using modern methods of genetics and bioinformatics, including genome sequencing on the Illumina sequencing platform.

The conclusions are consistent with the evidence and arguments presented.

The references are very relevant, including some relevant authors experience in the field.

I recommend some small corrections.

1.     Chap. 2.1. Ethics- Institutional Comitee and code/nr should be provided

2.     References should follow more carefully the MDPI Guide for authors. Some references contain integral name of journal, some others gave just first author et al (eg ref 28)

In my opinion, with a more detailed Introduction and Discussions sections, including  you may try to send your article to Nature group.

Success! 

Author Response

(The authors gave the same response as above.)

Reviewer 3 Report

Comments and Suggestions for Authors

In this research, the authors performed a genome-wide association study (GWAS) in buffalo for several milk production traits including 305-day milk yield, peak milk yield, total protein yield and total milk fat yield. The study was done on 120 individuals from 4 breeds. The sample size was relatively small for a GWAS, but sufficient to ensure enough statistical power without compromising the results. The fact that whole genome sequencing was done for genotyping is a plus for this study and, considering the costs of this technology, may justify the smaller sample size.

The employed methodology for WGS data analysis and for performing the GWAS was generally suitable and well described in the manuscript. However, there are several crucial amendments that need to be performed to the methodology, which are likely to significantly change the results. Because of this, the Results, Discussions and Conclusion sections will need to be revised accordingly.

The main issue is that the genotype filtering criteria are too stringent. Going down from an initial set of 2,208,174 genetic markers to a final set of only 18,352 filtered markers means that the authors have, effectively, thrown away more than 99% of the genetic data. While this can be a sign of problematic DNA sequencing, the authors should carefully consider their data analysis approach, given that their goal is to perform GWAS. Such a stringent quality filtering practically negates the advantages brought by the use of WGS, whereas the number of 18,352 is even lower than typical numbers of markers (after quality control) which are used in GWAS studies based on microarray data (which normally start with 50-60,000 markers before filtering). The authors need to slightly relax the quality control criteria for genotype markers in PLINK and use the new set of filtered markers for all of the successive stages of the study, including the population structure analysis.

Based on the reduced set of filtered markers (18K) initially used, there doesn’t seem to be a sign of significant population structure effects (the PCA clusters are not clearly separated). This is surprising, given the fact that the animal sample includes individuals from 4 breeds. This may well be a consequence of the stringent quality control criteria in PLINK and the relatively small number of markers used for the analysis. The authors should carefully reinterpret the results after relaxing the quality control filters. In case there will be stronger evidence for population structure effects, then the authors should also include a suitable number of principal components (PCs) as fixed effects in the GWAS implementation (in TASSEL). The number of PCs can be decided by creating a Scree plot, or by using the number of clusters that are found by Admixture. To test whether these values match, the authors should also create a Scree plot and include it in the main article.

The p-value significance threshold was set to 1/N, where N is the number of filtered markers. This strange choice seems to be ad-hoc and is difficult to justify. Typical use of Bonferroni correction is based on a threshold of 0.05/N. Alternatively, if this is too stringent, FDR adjustment can be used e.g. with the Benjamini-Hochberg method (while setting FDR to a value of 5%). The authors should use one of these standard approaches for the p-value significance in their study.

Finally, the authors should also include two additional phenotypes: total fat percentage (FP) and total protein percentage (PP). These are very easy to compute from MY, based on FY and PY, respectively. It would be interesting to see if genes like DGAT1 (which are famously found in GWAS associated with FP and/or PP in cattle) would also come up in this study based on buffalo individuals. If so, this would significantly increase the relevance of this study’s results.

The quality of English is good enough to understand the text, but minor issues are present and should be solved. Punctuation is used inconsistently across the text. Authors should make sure that every comma and dot is followed by a space (and that there is no space before them). Similarly, there should be a space before every open parenthesis and squared bracket. All the gene names should be italicized. When using the expression “et al.” for referencing, there should always be a dot after the word “al”.

Other specific remarks:

L30: The first word should be capitalized
L73: “Lslam” -> “Islam”
L85: At the end of the Introduction there should be one or two sentences that summarize the concrete objectives of this research.
L180: A reference should be included for dbSNP
L187: A reference should be included for KEGG
L200: The authors should include implementation details for the methods described in Section 2.9 (which programming language was used, which version, which software packages etc.)
L259: The Methodology selection does not explain how the cross-validation analysis from Figure 3 was performed. If this comes from Admixture, then this needs to be better clarified
L325: The minus sign should be on the same line with the number that it precedes
L326: “in five KEGG_A_class” -> “in five KEGG_A_class pathways”
L364: In order to improve visibility and ease of understanding, horizontal bars should be added to Table 3 in order to better separate the groups of 3 genotypes for each gene
L389: This paragraph suggest that the employed methodology included adjustment for population structure. However, the described methodology does not seem to suggest that this was actually implemented in TASSEL (and, if so, how it was done). This aspect needs to be clarified
L450: “association analysis on” -> “association analysis was performed on”
L461: Some suggestions for future work should be included near the end of the Conclusions section

Author Response

(The authors gave the same response as above.)

Round 2

Reviewer 3 Report

Comments and Suggestions for Authors

The revised version addresses all of the points from the original review. It is great to see that DGAT1 came up associated with FP in this study as well. This further indicates the validity of this study's methodology and increases the impact of its results. Overall, the revised manuscript is a considerable improvement over the originally submitted version and no further issues have been found.